# The Impact of the COVID-19 Pandemic Lockdown on Pediatric Infections—A Single-Center Retrospective Study

**DOI:** 10.3390/microorganisms10010178

**Published:** 2022-01-14

**Authors:** Magdalena Grochowska, Dominika Ambrożej, Aneta Wachnik, Urszula Demkow, Edyta Podsiadły, Wojciech Feleszko

**Affiliations:** 1Department of Pediatric Pneumonology and Allergy, Medical University of Warsaw, 02-091 Warsaw, Poland; magdalena.grochowska01@gmail.com (M.G.); dominika.ambrozej@wum.edu.pl (D.A.); 2Doctoral School, Medical University of Warsaw, 02-091 Warsaw, Poland; 3Department of Laboratory Diagnostics and Clinical Immunology of the Developmental Age, Medical University of Warsaw, 02-091 Warsaw, Poland; a.wachnik@interia.pl (A.W.); urszula.demkow@uckwum.pl (U.D.); edyta.podsiadly@uckwum.pl (E.P.); 4Centre for Preclinical Research, Department of Pharmaceutical Microbiology, Faculty of Pharmacy, Medical University of Warsaw, 02-091 Warsaw, Poland

**Keywords:** pediatrics, SARS-CoV-2, hospitalizations, respiratory tract infection, respiratory viruses, RSV, epidemiology

## Abstract

Since the SARS-CoV-2 outbreak, many countries have introduced measures to limit the transmission. The data based on ICD-10 codes of lower respiratory tract infections and microbiological analysis of respiratory and gastrointestinal infections were collected. The retrospective five-year analysis of the medical records revealed a substantial decrease in respiratory tract infections during the pandemic year (from April 2020 to March 2021). We noted an 81% decline in the LRTI-associated hospital admissions based on the ICD-10 analysis (from a mean of 1170 admissions per year in the previous four years to 225 admissions between April 2020 through March 2021). According to microbiological analysis, there were 100%, 99%, 87%, and 47% drops in influenza virus, respiratory syncytial virus, rotavirus, and norovirus cases reported respectively during the pandemic season until April 2021 in comparison to pre-pandemic years. However, the prevalence of gastrointestinal bacterial infections was stable. Moreover, in August 2021, an unexpected rise in RSV-positive cases was observed. The measures applied during the COVID-19 pandemic turned out to be effective but also had a substantial contribution to the so-far stable epidemiological situation of seasonal infections.

## 1. Introduction

It has been about two years since the outbreak of the COVID-19 pandemic [1]. The data reflecting the level of spread of the SARS-CoV-2 (as of 29 December 2021) show as follows: number of confirmed cases worldwide—281,808,270; number of deaths in the world—5,411,759; number of confirmed cases in Poland—4,080,282; number of deaths in Poland—95,707; case fatality rate (CFR) worldwide—1.92; CFR in Poland—2.35 [2]. The clinical manifestation of COVID-19 patients mostly incorporates headache, loss of smell, fever, cough, or myalgia. However, the prevalence of symptoms significantly varies according to age and sex [3,4]. In the pediatric population, the most common course of COVID-19 is usually mild to moderate and presents as fever, cough, diarrhea or vomiting [5]. In response to the rapid escalation of infection cases, many countries have introduced measures to limit the transmission of the new coronavirus, from the recommendation of social distancing and wearing masks to strict lockdown [1]. The pandemic measures provided by the Polish government varied overtime of the study (from the beginning of the pandemic to August 2021) and included mandatory use of face masks (excluding children under 5 years old) and 1.5 m minimum distance in public spaces, total or at least partial closure of kindergartens/nurseries, primary schools, secondary schools and higher education. The restrictions also concerned the restaurants, sport centers and other services from many sectors [6,7,8]. Our aim was to investigate how the SARS-CoV-2 pandemic has changed the hospitalizations frequency associated with lower respiratory tract infections (LRTI) in children and the incidence of the most common pediatric pathogens detected from hospitalized patients by microbiological analysis.

## 2. Materials and Methods

We retrospectively analyzed the results of the routine microbiology tests and the electronic health records of children hospitalized in the Medical University Children’s Hospital in Warsaw, one of the two largest pediatric hospitals in the capital of Poland with more than 30,000 annual admissions. This study received an exemption from ethics review at the Medical University of Warsaw (AKBE 138/2021) as it used aggregate prevalence data only, without any patient or clinician identifiers. Data were collected and processed using Microsoft Excel 2019.

Individual medical records involving LRTI-associated hospital admissions (based on ICD-10 codes): (1) viral infections, including bronchiolitis, (2) influenza and flu-like infections, and (3) bacterial infections; and detection rates were collected for the following pathogens: (A) respiratory viruses (*Influenza A, B, A*/*H1N1* viruses, and RSV), (B) gastrointestinal (GI) viruses (Rotavirus and Norovirus), (C) gastrointestinal bacteria (*Salmonella* spp., *Campylobacter* spp., *Yersinia* spp., and *Enterohemorrhagic Escherichia coli* (EHEC)), and (D) SARS-CoV-2 and compared between April 2020 to March 2021, and four corresponding previous seasons (2016–2020) with the addition of the summer 2021 period to demonstrate the peak in RSV-positive infections. Children of all ages were, regardless of their length of stay, included in our complete-case analysis.

### 2.1. Data Mining Based on ICD-10

For data collection on hospitalization rates, the electronic health record system CGM CLININET (CompuGroup Medical, Lublin, Poland) of the Children’s Hospital of the Medical University of Warsaw was used.

All patients, meeting the following criteria, were enrolled:The date of hospitalization in one of the following time periods: 1 April 2016–31 March 2017; 1 April 2017–31 March 2018; 1 April 2018–31 March 2019; 1 April 2019–31 March 2020; 1 April 2020–31 March 2021.The cause of hospitalization (based on clinical diagnosis and the ICD-10 code) due to one of the clinical subgroups, including lower respiratory tract infections: (1) viral infections (J12 (.0, .1, .2, .8, .9); J20 (.3, .4, .5, .6, .7, .8, .9); J21 (.0, .8, .9); J40], including bronchiolitis (J21 (.0, .8, .9)); (2) influenza or flu-like infection (J10 (.0, .1, .8); J11 (.0, .1, .8)); (3) bacterial infections (J13; J14; J15 (.0, .1, .2, .3, .4, .5, .6, .7, .8, .9); J16 (.0, .8); J18 (.0, .1, .2, .8, .9); J20 (.0, .1., 2)).

### 2.2. Microbiological Analysis

#### 2.2.1. SARS-CoV-2

Nasopharynx swabs were obtained from examined subjects. Total RNA was extracted from 140 μL specimen with a manual centrifuge column isolation kit (Viral RNA Isolation Kit, ZJ Bio-Tech C, Shanghai, China) according to manufacturer’s instruction. Real-time RT-PCR (Liferiver Novel Coronavirus (2019-nCoV) Real-Time Multiplex RT-PCR Kit, Shanghai ZJ Bio-Tech C, Shanghai, China) was performed upon CFX96™ Real-Time PCR Detection System (Bio-Rad, Hercules, CA, USA). The test detects 3 target genes: SARS-CoV-2 gene E, gene N, ORF1ab. The limit of detection is 1 × 10^3^ copies/mL. The specificity of the test is 98.1%, analytical sensitivity 1 × 10^3^ copies/mL. The protocol was performed according to the manufacturer’s instructions. Real-time RT-PCR was performed under the following conditions: 45 °C for 10 min and 95 °C for 15 min, followed by 45 cycles of amplification at 95 °C for 15 s and 60 °C for 1 min. Criteria for judging results: CT value < 43 positive; 43 ≤ CT value ≤ 45 suspicious positive and ≥45 negative. The positive should meet at least 2 genes detected. Internal control was added to the extraction mixture in the isolation stage to monitor the whole process. Positive and negative control was included in each run of amplification.

#### 2.2.2. RSV and Influenza A/B

The nasopharyngeal swabs were collected at admission to the hospital. The specimen was tested for the presence of RSV and *Influenza A/**B* virus using immune-chromatographic assay according to the manufacturer’s instructions (Alere, Scarborough, ME, USA). Some samples were tested with qualitative real-time PCR test Xpert^®^ Xpress Flu/RSV (Cepheid, Sunnyvale, CA, USA) according to the manufacturer’s instructions.

#### 2.2.3. Enteric Viruses

Enteric viruses: Rotavirus, Norovirus GI/GII were investigated by the immunochromatographic method according to the manufacturer’s instructions (NADAL Rotavirus, Nal von Minden and Simple Norovirus, Operon, Zaragoza, Spain).

#### 2.2.4. EHEC

The presence of stx I, stx II, and eae genes in *E. coli* strains isolated from stool samples was determined by real-time PCR according to EU protocols—identification and characterization of Verotoxin—producing *Escherichia coli* (VTEC) by Real-Time PCR amplification of the main virulence genes and genes associated with serogrups mainly associated with severe human infection. EU-RL VTEC Method 02 Rev of 5 March 2013.

#### 2.2.5. Salmonella, Yersinia, Campylobacter

Stool samples tested for the presence of *Salmonella, Yersinia* were plated on five different media: Salmonella Shigella Agar, Selenite -F Broth, MacConkey, Yersinia Selective Agar (Oxoid, Basingstoke, UK) and incubated under aerobic conditions at 37 °C. Stool samples directed to *Campylobacter* were plated on CASA Chromogenic Medium (BioMerieux, Craponne, France) and incubated under a microaerobic atmosphere at 42 °C. After initial sample processing, colonies were selected for identification using a Microflex LT mass and the MBT Compass IVD Biotyper software (Bruker Daltonics, Bremen, Germany). A score > 1.9 was considered a reliable identification at the species level.

## 3. Results

### 3.1. General Hospital Admissions

Following previous data, there was a noticeable seasonal distribution among the respiratory and GI viral infections with their peak during winter-spring months before the pandemic [9,10]. The number of total hospital admissions decreased by 25% during the pandemic season from a mean of 35,658 cases annually in the pre-pandemic years to 26,621 (−25%; −2,7 SD) cases in 2020/2021 (Table 1).

### 3.2. Lower Respiratory Tract Infections-Associated Hospital Admissions Frequency

Among registered hospitalizations, those resulting in LRTI-associated hospitalization accounted for a mean 3.3% (1170/35,658) of all admissions annually (with a total of 4905 cases between 2016–2021) and were subsequently classified, according to ICD-10 codes, into the following clinical categories: (1) viral infections, including bronchiolitis, (2) influenza and flu-like infections, and (3) bacterial infections. During the 2016–2020 seasons, the cumulative annual LRTI-associated hospital admission frequency remained steady (mean ± SD: 1170 ± 141 cases). In contrast, this trend was not reflected during the last 2020/2021 season when the total LRTI admission rate sharply dropped to 225 hospitalized patients (−81%; −6.7 SD).

### 3.3. Microbiological Analysis of Pathogens from Hospitalized Patients

The microbiological analysis revealed a significant decrease in respiratory and gastrointestinal viral infections in March 2020 after the lockdown announcement (red arrow in Figure 1). The number of positive tests for respiratory viruses (excluding SARS-CoV-2 infections) dropped by more than 99% (from mean 291 in 2016–2020 to 1 in 2020/2021). A similar tendency was observed for GI viruses with a 66% reduction, whereas a slightly increasing trend in viral gastrointestinal infections was noted in May 2020 after the opening of nurseries and kindergartens (green arrow in Figure 1). However, this trend was not observed for GI bacteria, with a mean yearly detection at 110 isolates in 2016–2020 and 95 isolates in 2020/2021, showing a steady trend over the past five years. Interestingly, no single case of flu and only one case of RSV was noted until a recent spike in RSV-positive infections in August 2021.

## 4. Discussion

The study demonstrates that a regional epidemiological situation remained stable before the SARS-CoV-2 pandemic, with a plummet of overall hospital admission rates quickly upon the introduction of lockdown in Poland. The decrease was most pronounced in rates of respiratory infections. Atypically, the RSV bronchiolitis season arrived early in 2021, confirming the latest global reports [11]. The incidence of GI viruses, predominantly feco-orally transmitted and requiring direct human contact, was reduced. However, the prevalence of bacterial GI pathogens, which mainly have food-associated transmission routes, did not fall significantly, similarly to previously reported results [12].

Besides the drastic drop in social interactions and gatherings, these associations may have been related to a fear of mandatory screening testing at admission and a fear of hospitals as the likely sources of the SARS-CoV-2 infection [13]. Interestingly, the opening of daycare centers only (and continuation of the lockdown for the adult population) was not followed by the marked increase in respiratory or gastrointestinal viral infections in children. The fact that in our research, we compared the full pandemic season paralleling with previous years’ periods is highly illustrative of the epidemiological situation.

To our knowledge, this is one of the first reports addressing the current pediatric epidemiological situation regarding not only the respiratory viruses but with other most common infections in a hospital setting in Europe [14,15,16]. The novelty of this paper is in the inclusion of non-respiratory infections covering different gastrointestinal pathogens. However, the present study has some limitations. Firstly, it is a single-center study. Secondly, the sampling bias cannot be excluded as our analysis reflects the group of moderate and severe infections in children who needed to be assessed in a hospital setting. Additionally, the retrospective analyses have often been associated with an increased risk of bias, and it is not possible to definitively prove a cause-effect relationship of the drown conclusions. We are aware of the fact that the proportion of children has changed since the start of the pandemic, and the reduced attendance to daycare might have affected the observed results. On the other hand, our study has some important strengths. Through a combination of medical records and microbiological data, we provide a complex view of how the SARS-CoV-2 virus has affected the work of one of the regional pediatric hospitals in Europe. It informs further prospective research using more rigorous scientific methods during the COVID-19 pandemic.

## 5. Conclusions

Our research illustrates that lockdown and social distancing reflect not only in the commonly acknowledged decline of reported SARS-CoV-2 cases but also in the distribution of other common infections in children. Our findings show the effectiveness of restrictions in preventing LRTIs and other common infections among pediatric patients and may imply applicability and future stewardship of public health strategies towards SARS-CoV-2 and other pathogens.

## Figures and Tables

**Figure 1 microorganisms-10-00178-f001:**
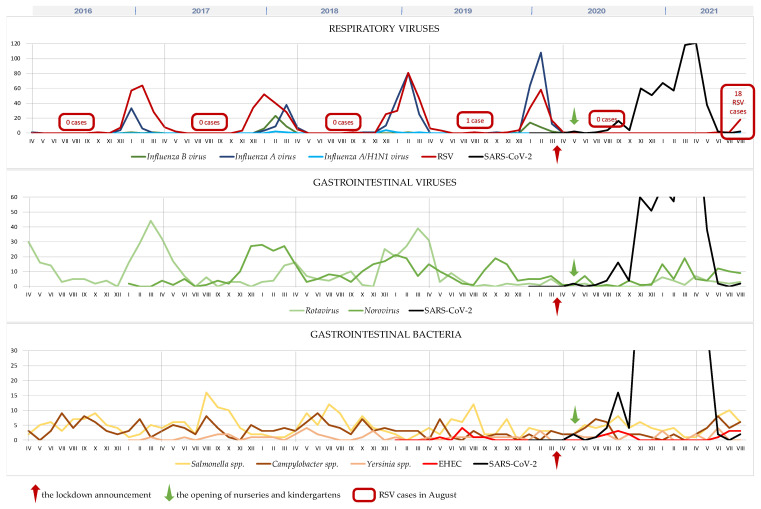
Monthly distribution of pathogens analyzed by Department of Microbiology during 2016–2021 seasons. RSV—respiratory syncytial virus, EHEC—*Enterohemorrhagic Escherichia coli*.

**Table 1 microorganisms-10-00178-t001:** Annual summary of infections during the 2016–2021 seasons at the Children’s Hospital of the Medical University of Warsaw. LTRIs—lower respiratory tract infections, SD—standard deviation, RSV—respiratory syncytial virus, EHEC—*Enterohemorrhagic Escherichia coli*, NA—not applicable.

	Seasons	2016/17	2017/18	2018/19	2019/20	Mean	2020/21	Change in 2020/21
	(in Periods from April to March)					(±SD)		% of Mean Value (as per SD) in 2016–2020
						2016–2020		
**ICD-10 Results**								
1	Respiratory viral infections (J12; J20 (apart from .1, .2); J21; J40)	462	414	406	412	424	102	−76%
						(±26)		(−12.4 SD)
	(including Bronchiolitis (J21)	161	202	184	261	202	24	−88%
						(±43)		(−4.2 SD)
2	Flu (J10; J11)	65	116	136	127	111	0	−100%
						(±32)		(−3.5 SD)
3	Respiratory bacterial infections (J13; J14; J15; J16; J18; J20 (.0, .1, .2 only)	602	812	661	467	636 (±143)	123	−81%
								(−3.6 SD)
	**Total LRTIs (ICD-10)**	1129	1342	1203	1006	1170 (±141)	225	−81%
								(−6.7 SD)
**Microbiological Analysis Results**								
A	Influenza viruses (combined)	47	91	179	210	132	0	−100%
						(±76)		(−1.7 SD)
	RSV	157	167	188	124	159	1	−99%
						(±27)		(−5.9 SD)
B	Rotavirus	168	89	161	59	119	16	−87%
						(±54)		(−1.9 SD)
	Norovirus	2	133	130	100	91	55	−40%
						(±61)		(−0.6 SD)
C	*Salmonella* spp	56	65	60	52	58	48	−18%
						(±6)		(−1.8 SD)
	*Campylobacter* spp	49	42	52	21	41	28	−32%
						(±14)		(−0.9 SD)
	*Yersinia* spp	1	9	14	11	9	11	+26%
						(±6)		(+0.4 SD)
	EHEC	NA	NA	NA	7	7	8	+14%
D	SARS-CoV-2	NA	NA	NA	0	0	380	NA
	Total Microbiological Pathogens	480	596	791	587	614	547	−11%
						(±130)		(−0.5 SD)
Total of General Hospital Admissions		31,084	35,597	38,611	37,339	35,658 (±3290)	26,621	−25%
								(−2.7 SD)

## Data Availability

Data are available upon request.

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
