# Peer review of "The Impact of the COVID-19 Pandemic Lockdown on Pediatric Infections—A Single-Center Retrospective Study"

_microorganisms, 2022, doi:10.3390/microorganisms10010178_

Round 1

Reviewer 1 Report

  • The authors mentioned there are measures to limit the transmission of SRAS-CoV-2 in many countries, including social distancing, wearing masks to stick lockdown etc. And the authors retrieve data from children's hospitals in Poland for data analysis. -
  • However, the authors did not mention the pandemic measures for Warsaw in Poland, which maybe related to this study.
  • Also, no ages of the children patients were mentioned, are they infants, school children ?
  • Are the school closed during the mentioned period? Did the children required to wear masks when they go outdoors/schools (if the schools are not closed).
  • Are all the children short-term patients in the hospital or some of them are long-term? 
  • All of these are very important epidemiology information for " the impact of COVID-19 pandemic lockdown on Pediatric infections"  The authors should consider collecting these information and include in their analysis , in discussion and in supplementary materials

- italic for all species names

Author Response

Professor Wojciech Feleszko

Medical University of Warsaw,

02–091 Warsaw, Poland

[email protected]

Dear Reviewer 1,

Microorganisms

Re: [Microorganisms] Manuscript ID: microorganisms-1534183 - Major Revisions

The Impact of COVID-19 Pandemic Lockdown on Pediatric Infections

Dear Sir/Madam,

On behalf of my co-authors, we would like to thank you for your consideration of our manuscript. We have revised it in response to the comments (changes have been added by using the "Track Changes" function in Microsoft Word), and the authors have all reviewed and agreed on the revisions made in response to the questions and comments below.

We remain at your disposal for any questions you may have.

Best regards,

Wojciech Feleszko, MD, PhD

Response to Reviewer 1 Comments

Point 1: The authors mentioned there are measures to limit the transmission of SARS-CoV-2 in many countries, including social distancing, wearing masks to stick lockdown etc. And the authors retrieve data from children's hospitals in Poland for data analysis. However, the authors did not mention the pandemic measures for Warsaw in Poland, which maybe related to this study.

Response 1: Thank you kindly for your insightful comment. We have incorporated to the manuscript a few sentences about the restrictions applied in Poland to limit the spread of SARS-CoV-2, basing on data from European Centre for Disease Prevention and Control and Polish governmental websites. (P1-2; L42-48)

Point 2: Also, no ages of the children patients were mentioned, are they infants, school children?

Response 2: We thank the reviewer for this question. We decided to omit the patients’ age due to the partial lack of data, making summary and conclusions difficult to provide.

Point 3: Are the school closed during the mentioned period? Did the children required to wear masks when they go outdoors/schools (if the schools are not closed)?

Response 3: Thank you kindly for your comment. We have followed the suggestion and we added some information to the Introduction. (P1; L42-47)

Here is the data on the above issues according to rules applied in Poland from March 2020 to August 2021.

The nurseries / kindergartens closing periods: 12.03.2020-6.05.2020 (total closure) and 27.03.2021-18.04.2021 (partial closure).

The primary / secondary schools closing periods: 12.03.2020-24.05.2020 (total closure); 25.05.2020-30.06.2020 (partial closure); 24.10.2020-9.11.2020 (partial closure); 10.11.2020-17.01.2021 (total closure); 18.01.2021-19.03.2021 (partial closure); 20.03.2021-25.04.2021 (total closure); 26.04.2021-28.05.2021 (partial closure).

Rules for wearing masks in children: According to governmental act from 6th May 2021, poz.861, wearing masks is obligatory for children in Poland, except for 4 years old children and younger.

References:

https://www.ecdc.europa.eu/en/publications-data/download-data-response-measures-covid-19

Point 4: Are all the children short-term patients in the hospital or some of them are long-term?

Response 4: We thank the reviewer for this comment and agree with his opinion about the importance of this information. However, we do not have complete data on this subject, so we decided to omit it in this analysis. Instead, we incorporated to the methodology section the sentence that we did not have any exclusion criteria due to age or length of hospital stay. (P2; L68-69)

Point 5: Italic for all species names.

Response 5: We thank the reviewer for their valuable input. As the reviewer had suggested, we corrected the names according to the comment. (P2-5; L: 63, 64, 103, 114, 121, 144, 145, Table 1., 170)

Thank you again for pointing out all these errors. We have corrected them all.

We wish to thank the reviewer for his thorough commentary and we hope that you will find our manuscript of a better form, more suitable for publication.

Yours faithfully,

Wojciech Feleszko MD, PhD

Reviewer 2 Report

Reviewer Comments to Author(s)

I read carefully this manuscript „The Impact of COVID-19 Pandemic Lockdown on Pediatric Infections” sent to the scientific journal „Microorganisms” (IF: 4.128). The idea of the manuscript is very good. However, I have comments and recommendations to this manuscript.   

(1.) Title: I recommend you change the title with a new title: "The Impact of COVID-19 Pandemic Lockdown on Pediatric Infections: A Single-Center Retrospective Study from Poland".   

(2.) Introduction / Background: I would like to raise the quality of this section. In this regard, I recommend to write information for pandemic of the COVID-19 / SARS-CoV-2 Infection (data to date: December 27, 2021): - number of confirmed cases worldwide; - number of deaths in the world; - number of confirmed cases in your country; - number of deaths in your country; - case fatality rate (CFR) worldwide; - case fatality rate (CFR) in your country.   

(3.) Introduction / Background: I think that it is highly recommended you write 1-2 sentences for the main clinical characteristics in patients with COVID-19 / SARS-CoV-2 Infection. In this regard, I highly recommend to add (write) the following scientific publications to this section (and add to section "References"): - [PMID: 32109013]; - [PMID: 32352202]; - [PMID: 33054699]; - [PMID: 32077115]; - [PMID: 32031570]; - [PMID: 32298988].   

(4.) Materials & Methods: I recommend you write a new subsection "Inclusion and Exclusion Criteria" (3-4 sentences).  

(5.) Results, Figure 1: I recommend you to correct / revise part "Respiratory Viruses" you sort out the pathogens in the following order: - "Influenza A virus"; - "Influenza A/H1N1pdm09 Virus"; - "Influenza B Virus"; - "Respiratory Syncytial Virus, RSV"; - "SARS-CoV-2".  

(6.) Results, Figure 1: I recommend you can change the current "gray" color of basic numbers and lines with new color "black". This will improve the quality of the figure and will make the figure more visible and readable for readers.  

Author Response

Professor Wojciech Feleszko

Medical University of Warsaw,

02–091 Warsaw, Poland

[email protected]

Dear Reviewer 2,

Microorganisms

Re: [Microorganisms] Manuscript ID: microorganisms-1534183 - Major Revisions

The Impact of COVID-19 Pandemic Lockdown on Pediatric Infections

Dear Sir/Madam,

On behalf of my co-authors, we would like to thank you for your consideration of our manuscript. We have revised it in response to the comments (changes have been added by using the "Track Changes" function in Microsoft Word), and the authors have all reviewed and agreed on the revisions made in response to the questions and comments below.

We remain at your disposal for any questions you may have.

Best regards,

Wojciech Feleszko, MD, PhD

Response to Reviewer 2 Comments

Point 1. Title: I recommend you change the title with a new title: "The Impact of COVID-19 Pandemic Lockdown on Pediatric Infections: A Single-Center Retrospective Study from Poland".  

Response 1: We thank for this comment. The title has been corrected accordingly. (P1; L3)

Point 2. Introduction / Background: I would like to raise the quality of this section. In this regard, I recommend to write information for pandemic of the COVID-19 / SARS-CoV-2 Infection (data to date: December 27, 2021): - number of confirmed cases worldwide; - number of deaths in the world; - number of confirmed cases in your country; - number of deaths in your country; - case fatality rate (CFR) worldwide; - case fatality rate (CFR) in your country.  

Response 2: We thank the reviewer for this valuable comment. Indeed, this information is very helpful to highlight the epidemiological situation worldwide and in Poland. We have added the suggested sentence to the manuscript’s Introduction, basing on World Health Organization data. (P1; L32-35)

Point 3. Introduction / Background: I think that it is highly recommended you write 1-2 sentences for the main clinical characteristics in patients with COVID-19 / SARS-CoV-2 Infection. In this regard, I highly recommend to add (write) the following scientific publications to this section (and add to section "References"): - [PMID: 32109013]; - [PMID: 32352202]; - [PMID: 33054699]; - [PMID: 32077115]; - [PMID: 32031570]; - [PMID: 32298988].  

Response 3: Thank you kindly for your insightful comment. We have followed the suggestion and we modified the Introduction as proposed by the reviewer. (P1; L36-40)

Point 4. Materials & Methods: I recommend you write a new subsection "Inclusion and Exclusion Criteria" (3-4 sentences). 

Response 4: Thank you for the advice. We completed the methodology section by adding an information that our analysis was a complete-case one and we did not have any exclusion criteria due to age or length of hospital stay. (P2; L68-69)

Point 5. Results, Figure 1: I recommend you to correct / revise part "Respiratory Viruses" you sort out the pathogens in the following order: - "Influenza A virus"; - "Influenza A/H1N1 Virus"; - "Influenza B Virus"; - "Respiratory Syncytial Virus, RSV"; - "SARS-CoV-2". 

Response 5: We appreciate the reviewer’s vigilance and attention to detail. We wanted to correct the viruses’ names order, however, it resulted in data shift and the loss of the intended graphical effect. Thus, we decided to omit this suggested edition.

Point 6: Results, Figure 1: I recommend you can change the current "gray" color of basic numbers and lines with new color "black". This will improve the quality of the figure and will make the figure more visible and readable for readers.  

Response 6: We thank the reviewer for their feedback. As the reviewer had suggested, we decided to change the colors of the basic numbers, titles and names of the pathogens with “black”. (Figure 1.)

Thank you again for pointing out all these errors. We have corrected them all.

We wish to thank the reviewer for his thorough commentary and we hope that you will find our manuscript of a better form, more suitable for publication.

Yours faithfully,

Wojciech Feleszko MD, PhD

Round 2

Reviewer 1 Report

I have no more comments.

Reviewer 2 Report

I read carefully this revised manuscript „The Impact of COVID-19 Pandemic Lockdown on Pediatric Infections: A Single-Center Retrospective Study from Poland” sent to the scientific journal „Microorganisms” (IF: 4.128). I do not have comments and recommendations to this revised manuscript.